# Prevalence of Suicide Thoughts and Behaviours among Female Garment Workers Who Survived the Rana Plaza Collapse: An In-Depth Inquiry

**DOI:** 10.3390/ijerph18126326

**Published:** 2021-06-11

**Authors:** Humayun Kabir, Myfanwy Maple, Md Shahidul Islam, Kim Usher

**Affiliations:** 1School of Health, Faculty of Medicine and Health, University of New England, Armidale, NSW 2351, Australia; mmaple2@une.edu.au (M.M.); mislam27@une.edu.au (M.S.I.); kusher@une.edu.au (K.U.); 2Department of Sociology, University of Dhaka, Dhaka 1000, Bangladesh

**Keywords:** female garment workers, Rana Plaza collapse, trauma, suicide, Bangladesh, qualitative study

## Abstract

The Rana Plaza building collapse occurred on 24 April 2013 in Savar, near the capital city of Bangladesh, killing more than 1130 garment workers and injured about 2500, mostly females. Those who survived face ongoing challenges, including socio-cultural constraints, economic hardship, post-traumatic stress disorders (PTSD), depression, and critical health issues, which may lead to suicidal ideation and death. The aim of this article is to explore why and how female garment workers who survived the Rana Plaza collapse are now at risk of suicide thoughts and behaviours, and suicide death. Unstructured face-to-face interviews were held from April to July 2018 with 11 female garment workers who survived the Rana Plaza building collapse. Interviews continued until data saturation was reached. The interviews were tape-recorded and transcribed verbatim while simultaneously being translated into English from Bengali/Bangla. Transcripts were coded and thematically analysed. The study found that all participants were living with multiple risk factors of suicidal ideation (including low socio-economic status, poverty, social stigma, psychological distress, and trauma) which the participants directly linked to the collapse of the Rana Plaza building. Our analysis uses the three-step theory of suicide (3ST, Klonsky & May, 2015) to understand female Rana Plaza survivors’ suicide risk. Female survivors’ overall vulnerability requires urgent attention while taking the socio-cultural setting of Bangladesh into account. In addition, a lifelong caring system (combining financial security and free healthcare) needs to be initiated to accommodate the female survivors with mainstream society to avoid possible future suicides. They require long-term social and economic security and psychological support.

## 1. Introduction

A 27 year old man, Himu, set himself alight and died on 24 April 2019. Himu had worked as a volunteer during the Rana Plaza collapse on 24 April 2013 and rescued many victims and recovered as many bodies he could throughout the 17-day rescue operation. He later also worked for the rehabilitation of the injured victims [1,2]. It may be a coincidence that Himu died by suicide on the sixth anniversary of Rana Plaza building collapse. Yet, his suicide death highlights deep psychological traumas suffered as a result of the Rana Plaza collapse. This is in addition to the conditions under which those who survived the collapse live: living with economic hardship, physical and psychological ill-health, socio-cultural constraints, and trauma. Such vulnerabilities may result in suicidal thoughts and behaviours, which may lead to death.

Himu’s death is not the only suicide death reported from those involved in the building collapse. Less than one year after the collapse, on 24 January 2014, Salma, a 27-year-old female readymade garment (RMG) worker who survived the Rana Plaza collapse died by suicide. Police confirmed that Salma was suffering significant disability caused by the injuries sustained in the Rana Plaza collapse, and as a result this situation created psychological stigma and distress [3]. While the deaths of Himu and Salma have been reported in mainstream media and previous studies have reported on Rana Plaza survivors’ suicide deaths [4,5], the total number of suicides among survivors is unknown. The presentation of reliable data may reduce the reputation of the Bangladesh garment industry, resulting in changes to global supply chain through international brands’ concern for reputational damage, which will eventually affect RMG businesses on which the Bangladeshi economy is reliant. Therefore, there are pressures against reporting accurate suicide data in this industry [5,6]. 

Beyond Bangladesh, and across different disasters, survivors have been found to experience poor physical/mental health status and commonly suffer from post-traumatic stress disorders (PTSD), which can lead to suicide ideation and suicide attempts [7,8,9,10,11,12,13,14,15,16,17]. Among survivors, women are reported as being more vulnerable to PTSD as well as to suicide risks compared to men [10,12,14,15,16]. In Bangladesh, suicide rates are higher among women than men, which was primarily reported as being due to interpersonal and intimate partner violence, low socio-economic and educational status, economic dependence on men, lack of social support, and poor bonding with friends and family [18,19,20,21,22]. Adding to these interpersonal vulnerabilities, the socio-cultural structure of Bangladesh which severely inhibits female autonomy is considered to be a cause of higher suicide rates among women [22,23,24]. Based on this, it can be argued that there are significant gender differences in suicide thoughts and behaviours, and that suicide is a gendered phenomenon [25,26]. The current study focuses on how one garment factory disaster impacted female survivors and their experience of suicide thoughts and behaviours.

Considered the worst and deadliest collapse of a RMG building internationally [27,28,29], the Rana Plaza collapse killed approximately 1130 people and seriously injured another 2500 [4,30,31]. It is worth noting that due to the patriarchal ideologies that dominate and shape the economy and cultural contexts of Bangladesh [32,33], the Rana Plaza collapse impacted male and female workers in different ways [17,34,35]. For example, unlike males, female survivors are not permitted to undertake other labour roles that are available to them, including being rickshaw pullers/van drivers [5,17,33,35]. According to international reports, the deaths and injuries mainly involved young (aged 18–20 years) female garment workers [36,37,38]. Previous studies have explored elements of PTSD [39,40], rescue and emergency management during the collapse [29], and corporate social responsibility (CSR) [41,42] concerning the surviving workers following the collapse. Studies showed that psychological trauma and depression were prevalent among the Rana Plaza survivors [40,43], which could lead to suicidal ideation [4]. Most importantly, there have been other reports of suicide death linked to the collapse [5,17]. Overall, previous research reported that the number of deaths of the Rana Plaza workers could have been minimised if robust rescue policies and management existed and that the survivors are living with several vulnerabilities (e.g., depression, trauma, physical injuries, etc.) directly attributable to the lack of proper CSR initiatives, while suicide deaths among collapse survivors have also been reported. It is worth noting that after the Rana Plaza collapse incident, the existence of CSR initiatives in Bangladesh RMG sector has come into question [44]. Due to the nexus between state-business and corporate entities, workplace safety and conditions as well as CSR initiatives can be compromised. This results in regular disasters and workers’ rights violations during pre-disaster, disaster, and post-disaster periods in the Bangladesh garment sector [5,42]. While there is a focus on CSR in the RMG sector immediately during post-disaster periods, it is rarely implemented [31]. Thus, CSR remained as an ambiguous program for Western retailers, yet failed to benefit the Rana Plaza survivors [45]. However, there is a dearth in existing literature on how and why female survivors are at risk of suicide thoughts and behaviours.

Therefore, this article aims to identify whether there are distinctive factors that may increase the risk of suicidality in female Rana Plaza survivors, including ideation and risk of suicide death. To do so, qualitative in-depth interviews were undertaken with 11 female survivors of the Rana Plaza tragedy. As qualitative in-depth interviews are considered an effective data collection method from traumatised individuals, disaster survivors, and from the people who have attempted suicide [5,17,24,25,46], the study incorporated this approach. The syntheses of the results are presented using an established theory in suicidology, being the three-step theory (3ST) of suicide, to understand female Rana Plaza survivors’ suicide risk [47]. This model articulates the vulnerability conditions under which suicide can occur and how suicidal ideation can lead to suicide attempts and death. The model assists in demonstrating the study participants’ post-Rana Plaza collapse experiences (pain, hopelessness, and less/no connections with relatives, family members, husbands, neighbours, etc.) in relation to their suicidal thoughts, behaviours, and attempts. We present a modified version of the 3ST theory in Figure 1.

## 2. Materials and Methods

As part of a larger study on the health and wellbeing of the Rana Plaza building collapse survivors in Bangladesh (see [5]), this project sought to examine data from female participants with a specific focus on suicide-related experiences.

### 2.1. Design

A qualitative, in-depth interview design was applied in this study. This study was conducted and is reported in accordance with the consolidated criteria for reporting qualitative studies (COREQ) [48]. Approval was obtained to conduct this study from the University of New England Human Research Ethics Committee (approval number: HE17-277), on 20 December 2017. Details about methods and materials can be found at [5].

### 2.2. Study Population

Interviews were conducted from April to July 2018 through the use of a pre-developed interview guide by the first author. Before commencing the interviews, the participants were given a detailed description of the purpose and objectives of the study, and each of them signed the informed consent form. Interviews were conducted in Bangla/Bengali, the first language of the interviewees and the interviewer. For the larger study, the main topics covered in the interview were factors associated with the impacts of the collapse of Rana Plaza on participants, including an in-depth inquiry about the existing socio-demographic characteristics of the participants, reasons for going to work on the day of the collapse, experiences of social stigma after surviving the collapse, the existing status of mental health after the collapse, and ongoing physical health issues caused by the collapse. The topic interview schedule was pilot tested with two Rana Plaza survivors, who were not included in the final sample size. No adjustments were necessary following the piloting of the interview schedule. From the larger study, which constituted 17 participants (Female: 11, Male: 6), only female participants who reported suicidal ideation were included in this analysis. As presented, due to a traditionally patriarchal social structure, females experience less control and freedom. This has been reported as a contributor to higher suicide death rates for women than for men [22,23,24,33]. In addition, social stigma, religious issues, feelings of being a burden, being unskilled, domestic abuse, dowry issues, poverty, poor health status, other socio-cultural factors have all been identified as significant risk factors for suicide among Bangladeshi women [19,20,21,22,24]. Therefore, this analysis sought to identify female participants who reported suicide thoughts and behaviours only. Finally, a total of 11 female Rana Plaza survivors, aged 18 years and above were included in this analysis. Along with the broader topic issues, the interviewer used probing questions to better understand the role the Rana Plaza collapse has had on suicidal ideation and behaviours for these female participants.

### 2.3. Data Collection

All interviews were conducted by the first author, a current full-time PhD student in the University of New England (Australia), at a pre-arranged time in a mutually agreed location (in the two labour union offices, which is a culturally appropriate location for females to attend an interview with an un-related male researcher) in the Savar Upazila administrative area. The participants for the larger study were recruited purposively through using the snowball technique. Three contact persons (two of them were the leaders of Rana Plaza survivors’ association and the third person was the RMG trade union leader), assisted in recruitment. Additionally, the participants also shared information about the study with other survivors known to them. Prior to the interview commencing, the entrances to the trade union offices were closed to maintain privacy for the duration of the interview. Given the socio-cultural norms of Bangladesh which generally prohibit unaccompanied females being in the presence of non-related males, consideration was required in relation to the interview context. As the interview schedule did not include any personally sensitive information (i.e., discussions of intimate matters), it was decided that conducting the interviews in a public place (union office) without a female chaperone would be appropriate even though the interviewer was male. All interviews were audio-recorded, and field notes were taken. Precautions such as postponing the interviews, access to the nearby hospitals, enabling a psychologist to be connected via phone, and the presence of at least one RMG worker/trade union leader were taken to handle any traumatic situation that could arise during the interview period. The average duration of the interviews was 40–65 min. At the end of each interview, the interviewer shared the main topic of interest of the interview as well as field notes with the participants, which increased validity and highlighted any topics that were missed.

### 2.4. Data Processing and Analysis

Interviews were tape-recorded, transcribed verbatim, and then translated into English. Both English and Bangla scripts were then checked against the field notes as well as audio recordings and attested to by a research student at the University of New England (Australia) with expertise both in the Bangla and English languages, to ensure the accuracy of the data. Following transcription, data were reviewed by the authors HK and MM. Individuals preliminary points of interest (i.e., phrases, sentences, words, paragraphs, or even entire documents) were identified to interpret the data [5,46]. The transcripts were then thematically analysed using an interpretative lens to identify the themes that have been used previously in suicide related qualitative studies, c.f. [5,17,46]. The identification of themes was completed via three analytic rounds: (a) reading and rereading to be familiar with the content, (b) identification of the themes, and (c) applying the theme categories after being refined [49,50,51]. Through this process, unique units of meaning were identified, while short phrases were developed and then grouped under larger themes. To increase the validity and to highlight if any themes were overlooked, units and themes were cross checked. HK and MM jointly reviewed the analyses, discussed similar and dissimilar themes, and then finalised the themes and participants’ quotations under the suitable themes. Initially, six themes were agreed on from the data analyses. Due to repetitions and representing similar meaning, two themes were merged with the final four themes, presented in the ‘results’ section and Figure 1. All participants were given unique numbers/codes to protect their identity.

## 3. Results

Participants were comparatively young females (age range 19–40) with low educational backgrounds; only one participant (P03) had completed primary level of education (class 1–5) while the rest of the participants’ education levels were up to the primary school level. All were living with considerable physical limitations and mental health conditions, and faced socio-cultural and economic hardships, leading to increased vulnerability. Physical limitations and disability, including bone musculoskeletal, severe headaches, bodily pain, and breathing problems were commonly reported. In addition, mental ill-health including from depression, sleep disorders, trauma, nightmares, anxiety, and fear were identified. Socio-cultural constructions of work (i.e., women are not permitted to undertake certain employment roles), post-Rana Plaza workability (such as employers’ perceptions of limitations that might inhibit productivity), unemployment status, and disability all contributed to extreme poverty. All participants reported these vulnerabilities as directly related to the Rana Plaza collapse. Ten of the eleven participants reported that they did not have access to any free mental health care facilities at the time the interviews were conducted. The analysis of the interview data is illustrated in specific themes (Table 1). Each theme is explored below, utilising participants’ quotes to demonstrate the findings.

### 3.1. Gender, Socio-Economic, and Cultural Settings in Shaping Suicidal Ideation

Participants narrated how an incident like the Rana Plaza collapse could impact survivors in gendered ways. The socio-economic and cultural context of Bangladesh is an important feature that underlies the interviews, in that Bangladesh is a patriarchal society where women have little autonomy. The participants explained why they considered themselves to be more vulnerable compared to male survivors:


*P09: “I am struggling with poverty since the collapse because I am a woman. I am not allowed to do many jobs that a survived male worker can easily do. I have lost every hope to survive.”*


Not only did the participant experience a traumatic building collapse, but there is also no replacement industry in which she can regain employment. This sentiment was also shared by participant P02, who compared her experiences with her male colleagues who survived the collapse:


*P02: “Many of my male co-workers have become rickshaw puller, van driver, and day labour. I can neither be a rickshaw puller or van driver because our society will not allow/accept me to do that. I am fed up of struggling with my fate and want to end up with it.”*


In both of these examples, the participants end their statements with words that indicate hopelessness and helplessness. There is no hope to survive (P09) and being fed up with fate (P02) indicate the desperateness of their situations. This was common across the interviews with female survivors, where stories, such as that by P08, demonstrate that this hopelessness was a result of their experience after the collapse occurred, rather than the actual collapse of the building:


*P08: “After the collapse, I tried to find a job in the garment sector but the management of a factory did not hire me as they believed, being a female worker and having a history of injuries, I will not be able to fulfil the daily production target. Since then I am unemployed. I do not find any value of my life and I am greatly depressed…”*


Here, gender-segregated labour (where women are not expected to do all types of jobs) has undermined the post-Rana Plaza productive capacity of the female survivors. Participants compared themselves to male survivors, where female survivors explained they have been squeezed from alternative job opportunities. Unemployment has created more vulnerable conditions for the female survivors, including worsened socio-economic conditions and feelings of hopelessness and frustration with their lives. Given these survivors were unemployed, and often unable to obtain any independent financial security, they had no financial ability to access medical facilities for their ongoing physical and mental health concerns (note that very limited/no free access to healthcare facilities were available for the survivors at data collection time). As such their expected standard of living and the quality of life. Given these conditions, participants spoke of their wish to end their life, and/or their suicide attempts. 

### 3.2. Ongoing Physical Injuries/Disability, and Mental Trauma Linked to Suicidal Ideation

Along with the lack of employment options and resultant economic vulnerability, participants were still living with different injuries as a result of the collapse. Many had hands and/or legs amputated immediately after being rescued from the tragedy to stop further infections, thus leaving them with permanent physical disabilities. Even though significant time had passed since the collapse, continued medical issues were present among participants. For example, P11 survived despite still battling infection since the collapse and was still waiting for surgery to amputate one of her legs:


*P11: “Recently, doctors suggested me to be ready for an operation to amputate my right leg to stop further infections sourced from Rana Plaza collapse. I am worried about the rest part of my life without one leg.”*


Severe headaches and bodily pain were commonly experienced and to the extent that such pain was directly connected with thoughts of suicide, as P01 describes:


*P01: “Since the collapse, I have been living with severe headaches, shoulder, backbone, and leg pain… I cannot tolerate these pain and sometimes I want to kill myself.”*


In addition to significant, chronic, and intolerable pain, many participants spoke of the difficulties they experience with sleeping, which further compounds their experience of pain:


*P05: “I cannot sleep well due to the muscle pain sourced from the collapse of Rana Plaza. No free treatment is available now for us. Sometimes, the pain becomes unbearable which makes me upset about life and I wish to finish it.”*


Participants P02, P04, and P10 explained how their disability status, caused by the collapse, made them hopeless about their lives, their value as a woman, and their lives more generally:


*P02: “My husband is a day labour who earns BDT 350/400 ($3.50/4.50) per day. I always live with the fear if he does not feed me anymore and gets married to another woman. How long a man can feed a disabled woman?”*



*P04: “My husband left me last year due to my disability. I am not connected to anyone. I feel like I do not exist anymore and I do not find the value in living further.”*



*P10: “Life has become meaningless to me as I have become disabled to work to support myself and my family.”*


### 3.3. The Feeling of ‘Being a Burden’ Linked to Suicidal Ideation

Women who entered the RMG sector—who often came from low socio-economic and educational backgrounds—had to navigate several societal challenges, such as social stigma around female jobs in the RMG sector, moving from rural areas to urban cities for work, and being willing to live in an unknown environment. Unfortunately, when a sudden disaster like the collapse of Rana Plaza shuts down possible earnings, survivors, such as P03 and P01, had to rely on their poor families. Feelings of burdensomeness may lead to suicidal ideation:


*P03: “I started to work in the RMG sector as my father was too old to work as a day labour. My little siblings and parents relied on my income when I worked at Rana Plaza. As I have become unable to work after the collapse, my old father has started to work again. It is better to die than living as a burden, at least, my family will be freed from feeding me.”*



*P01: “Once, I was the main earning person and every member of my family used to respect me. Unfortunately, I have become a burden for them. Now, my family is suffering a lot to feed me and to buy medicines for me. I just cannot tolerate this painful life and I do not want this life anymore…”*


### 3.4. Lack of Social Support and Social Stigma Leading to Suicidal Ideation

The trauma of surviving the collapse at times isolated the participants within their own community. Participant P06 lost her own sister during the collapse but somehow managed to survive. She narrated how her family was driven away from the relatives and neighbours:


*P06: “I and my elder sister, who found as a spot dead, worked in the same factory at Rana Plaza. I cannot forget my sister’s dead face till today…I am living with trauma! My father died a few years ago. Our relatives now drive us away! Our family is treated as a ‘cursed family’ within the community...”*


While participant P06 described being stigmatised by others as being ‘cursed’, participant P07 had to move to a new living place from her previous residential area so that she was no longer recognised as a Rana Plaza survivor, even while knowing that she may be uncovered as a collapse survivor in her new place of residence:


*P07: “My husband died in 2009 in a road accident which led me to join in RMG work for survival. After being rescued from the collapse, I and my daughter were living as self-isolated in our previous rented house in Jamsing (a residential area in Savar, Dhaka) because we were not treated well by our neighbours. Our neighbours used to treat me as ALOKKHI (a euphemism for loss/bad luck) if anything unexpected happened in their daily lives. We shifted to a new area where I now have started a small grocery shop with the financial assistance received from different sources after the collapse. I hide the information that I am one of the Rana Plaza survivors.”*


Social stigma [treated as *ALOKKHI* (a euphemism for loss/bad luck), cursed, and the tendency to hide the identity of Rana Plaza collapse survivors] contributed to Rana Plaza survivors’ self-isolation and strained relationships with relatives and neighbours. Along with social stigma, lack of social support (misunderstanding with the family and relatives), losing friends, family members, and co-workers during the collapse (which remains as the most traumatic events in the memory of the survivors), as well as socially forced relocation from one residential area to an unknown area (to hide their identities) lead them to suicidal ideation. Participants describe a complex intersection between surviving a traumatic event that has resulted in long term physical pain, along with social exclusion and isolation resulting in mental anguish. Throughout the participant quotes, the helplessness of these women’s experiences can be felt in the choice of words to describe the desperateness of their situations. 

Overall, the themes articulated that the Rana Plaza female survivors were living with suicidal ideation (and expressed their wish to die by suicide) due to the patriarchal cultural settings and gendered construction of labour market, suffering from ongoing physical health problems and mental trauma directly linked to the collapse, and the changed context over the time when they had been treated as cursed, as a burden, and so forth.

## 4. Discussion

To our knowledge, this is the first qualitative study which has focused specifically on suicidal ideation as it is experienced by Rana Plaza female survivors. All 11 participants were found to be vulnerable to suicidal ideation, specifically related to the identified themes: gender, socio-economic, and cultural settings; ongoing physical injuries/disability and mental trauma, the feeling of ‘being a burden’, and a lack of social support and social stigma. Our study demonstrates that experiences of fear, hopelessness, depression, anxiety, trauma, social stigma, pain, and sleep disorders are common among the participants. These experiences, along with the cultural contexts in which Bangladeshi women live, lead participants to describe unresolved despair, with their narratives including their thoughts of suicide. Interestingly, while suicide is often conceptualised as a psychological pain–also known as psychache [52], the ways in which these participants spoke of their desire to end their lives related to an inability to see a way forth within a dominant culture that excluded them and their experiences, which indicates the more external focus of their suicide thinking than is reported in the dominant Western theorising about suicide. Western theorisations of suicides primarily deal with the capabilities and motivations to die by suicide [53,54], and focus on risk factors commonly reported in Western medically-oriented research [mood disorders, personality disorders, addiction to alcohol/drugs, sexual orientation (nonheterosexuality), self-harm attitudes, relationship breakdown], leading to suicidal ideation [25,46,55,56,57]. Meanwhile, the suicidal ideation reported among the study participants related to broader external factors including poor socio-economic status, lack of resources to meet basic needs (such as food and medicines), vulnerable physical and psychological health status directly linked to the Rana Plaza collapse, social stigma, and so on. Thus, these survivors’ suicide ideation needs to be understood more carefully, beyond the Western theorisations and constructions of suicide and suicide risk.

In previous studies reporting on the Rana Plaza building collapse [5,17,40], almost every participant was found to living with ongoing physical health issues, had very limited/no free healthcare facilities, tired of struggling with financial hardships, received inadequate financial compensations, and culturally stigmatised. Other researchers have reported a high prevalence of PTSD among the Rana Plaza survivors [40] and ongoing physical and psychological disorders, such as sleep disorders, anger, and depression [39]. In addition, Action Aid [36] tracked 1400 Rana Plaza survivors since 2013 and interviewed 200 of them, and reported that 27% of the survivors could not join in work due to ongoing poor mental health and 10.5% suffered the effects of trauma. While this work by Action Aid did not report on suicide among survivors, these experiences have previously been linked to suicide in other places. For example, Maple et al. [46], Im, Oh, and Suk [58], Woo and Postolache [56], and Greydanus and Calles [59] identified depression, trauma, losing friends/family and/or misunderstanding from them, sleep disturbance, mood disorder, hopelessness, and social isolation as major causes of suicidal ideation. According to Franklin et al. [57], individuals with any type of mental illness, serious or chronic physical illness, life stress, special population status (such as migrant), and access to lethal means (e.g., drugs) may be at risk for suicidal thoughts and behaviours (STBs). Kim, Jung-Choi, Jun, and Kawachi [60] demonstrated how socio-economic inequalities can lead to self-destructive behaviours including suicidal ideation among the South Korean people. In addition, Iemmi et al. [61] found that worse economic status, unemployment, and diminished wealth are associated with suicidal ideation.

Most of the risk factors mentioned in the aforementioned studies are present among the current study populations, which is a serious concern for the future. While suicide attempts are an underreported phenomenon in the developing countries such as Bangladesh [4], it is unknown how many Rana Plaza survivors attempted to die by suicide. The study also explores how social and cultural settings shape the economic activities of the female survivors differently. Unlike male survivors, they are excluded from some occupations (such as rickshaw puller/van driver) due to their gender, which limits their economic security. 

Suicide is considered to be a leading cause of death worldwide (ranked 15th) [61]. Suicidal behaviour is a global cause of death and disability with more than 800,000 deaths reported annually (one death every 40 s); 79% of suicidal behaviour cases occur in low-income and middle-income (LMIC) countries [62]. An estimated 10,000 suicide deaths occur in Bangladesh every year [63], although the exact number is likely higher given data and reporting challenges. Reasons for these deaths have focused broadly on the psychiatric disorder, physical disability, traumatic life events, economic hardship along with age, literacy rate, and place of residence (urban/rural) [18,64]. However, the focus on suicide in LMIC is considerably lower than in high-income countries, which contributes to a predominantly Western view theorising how and why suicide occurs while simultaneously not acknowledging the different risk groups.

To further explore the relevance of current suicide theorising, we applied the 3ST model [47] to our findings. The 3ST model uses three steps to understand how suicide can occur and how suicidal ideation can lead to suicide attempts and death. The first step refers to internalised hopelessness and pain. Across the sample, participants were living with psychological and physical pain described as a direct result of the Rana Plaza collapse. Joblessness, a lack of job options (caused by the socio-cultural constructions of work, e.g., women are not expected to be rickshaw pullers or van drivers), being fed up of struggling with the ongoing financial hardships, dependence on husbands and other family members, disability, tensions about ongoing infections which may lead them to amputate legs/hands, extreme bodily pain, trauma (for losing friends and family members during the collapse), or a lack of access to free healthcare facilities certainly forced the survivors to live in pain and hopelessness. The second step in the 3ST model focuses on connections to other people. The participants mentioned being isolated from others, as well as being alienated from both existing connections (e.g., spouses, families, and communities) due to the collapse as well as perceived as bringing ‘bad luck’ with them. The study shows that participants were less connected to their family members, relatives, and neighbours. The social stigma (*ALLOKHI* and treatment of as a cursed family) around the collapse of Rana Plaza absolutely fuels the continued painful lives of the survivors. Some of the participants expressed their psychological pain by saying that they always live in fear whether their husbands will leave them because of their disability. The husband of Participant 04 already left her due to her disability. Thus, the relationships between pain and connectedness not quantifiable; however, based on the survivors’ stories, it is assumable that their pain outweighs the already fragile connections with the husbands, relatives, friends, and neighbours. The third step of the model emphasises the capability of killing oneself. Almost all the participants showed their wishes to finish their lives by saying ‘want to end up with it (life)’, ‘do not find any value of life’, ‘want to kill myself’, ‘wish to finish it’, ‘do not exist anymore’, life has become meaningless’, ‘better to die’, ‘do not want this life anymore’, etc. These strong willingness clearly signpost that these survivors are capable of suicide attempts. The past suicide cases of the survivors have already proved this. However, it is hard to determine how many of the survivors will commit suicide in the future.

Our participants talked of the desire to end their lives; however, whether they would go on to act on these intentions is unknown due to the data collection methods. The step of moving from hopelessness into active intent to end one’s life has been the focus of much attention among suicidologists. Joiner [65] espouses that by nature (evolutionarily and biologically) people are wired to avoid injury, pain, and death. As a result, a clear move from suicide ideation to attempting to end one’s life is needed. According to O’Connor and Kirtley’s [53] IMV (integrated motivational-volitional) model, defeat and entrapment initiate the development of suicidal ideation. They also propose that entitled volitional moderators and a group of factors (such as access to the means of suicide, exposure to suicidal behaviours, capability for suicide (fearlessness about death and increased physical pain tolerance, planning, impulsivity, mental imagery, and past suicidal behaviour) trigger the transition between suicidal ideation and suicide attempts/suicide. We adapted the 3ST model to incorporate our findings, which suggested that these participants are at risk of ending their lives.

The propositions (e.g., pain, hopelessness, connectedness, and suicide capacity) of the 3ST model helped us to describe how the female Rana Plaza survivors were living with greater physical and psychological pain (directly linked to the building collapse), no hope for a better future (due to the inactive role of the factory owners, international brands/buyers, and local government), less/no connection with friends and family members (for ongoing poverty), and the greater likelihood to complete suicides in the near future. Finally, we argued that the socio-cultural contexts [which obstructed the participants from engaging in alternative job options and branded them as *ALOKKHI* (possessing social stigma)], the feeling of being a burden (e.g., transitioning from a breadwinner to a bread eater), ongoing health complaints, and a lack of appropriate support from the relevant agents may push them to suicide ideation-to-action, based on the framework proposed by Klonsky and May [47].

Given the vulnerable situations (which may lead them to suicidal deaths in future) of the female Rana Plaza survivors, this study strongly recommends that the relevant agents (such as BGMEA, international brands, GoB) should take urgent steps to articulate these survivors within the mainstream society by providing lifelong financial security and access to free healthcare facilities, and engaging them (those who are still capable of working) with doable as well as easy tasks in the RMG sector. Continued pressure on the RMG factory owners from international organisations, communities, and even from the clothing wearers is needed to ensure workplace safety in Bangladesh’s RMG sector. The findings of the study can be used to understand why and how disaster survivors in the similar kinds of circumstances within the South and Southeast Asian regions are vulnerable to suicide thoughts and behaviours.

### Limitations

Psychological components related to suicidal ideation (such as pain, hopelessness, lack of connectedness, depression, anxiety, fear, trauma, and social stigma) were found among the female RMG workers who participated in this research. The major limitations relate to the small number of participants, and lack of capacity to follow-up on the participants’ experiences beyond a one-off interview. There is no way to re-connect with these participants. In addition, exclusion of the male survivors limited the study findings in looking at similarities and dissimilarities (male vs. female survivors) across the themes identified. However, the data does shed light on the limitations of our current thinking about how suicide ideation occurs and the role of dominant cultural expectations that limit those already vulnerable from being able to find new ways of living beyond a significant trauma. 

## 5. Conclusions

This paper demonstrates the vulnerability of female Rana Plaza survivors. In Bangladesh, the gendered division of labour, social stigma, dependence on men/other family members and the feeling of ‘being a burden’, lack of bonding with relatives and neighbours, continuous struggle with financial hardships, ongoing physical health complications, limited or more commonly no access to free healthcare facilities, and the memory of the collapse lead these survivors to depression, PTSD, hopelessness, sleep disorders, psychological pain, trauma, and suicide. Other female survivors have already ended their own lives [3,5] and more recently a male survivor did so as well [1,2]. Thus, suicide prevention is an important consideration for those who survived the building collapse. Any prevention activities need to take into account the gendered nature of the Bangladeshi community, whereby female survivors are highly vulnerable to suicidal thoughts and behaviours due to the socio-cultural limitations on them, heightened by public stigma regarding the collapse and towards those who survived. While Western models of suicide prevention often focus on the individual risk factors, in this context social and economic factors also require attention. Specifically, while prominent theories of suicide explain the risk factors being primarily psychological, here external factors related to life circumstances, such as lack of basic needs, socio-economic structure, cultural barriers, social stigma, gendered construction of employment and society, coupled with ongoing physical and psychological from the factory building collapse, were found to be important central causes of suicidal ideation among the study participants. Thus, this study highlights important cultural distinctions when considering suicide in non-Western contexts. 

The GoB, BGMEA, and trade unions need to work together to address the social stigma around the Rana Plaza collapse survivors. Electronic and print media can play an active role in anti-stigma campaigns to assist in reducing pressures on these vulnerable members of the community. Further, as international brands and international organisations (such as the International Labour Organisation) hold significant power within the Bangladeshi economic and trade routes, pressure from these organisations to support Rana Plaza survivors is important, and this can extend to their families in the provision of healthcare, education, stable housing, and the like. These findings have applicability to other South and Southeast Asian RMG producing countries, whereby there is a need to focus on the wellbeing of the workers who are crucial to the success of the sector. This study is particularly important for international brands who rely on workers in RMG factories to take note of, particularly in the current climate with COVID-19 impacting the viability of RMG workers’ job security [66]. Such information can inform service delivery programs to support women who are culturally and economically bound and left feeling hopeless when their employment ceases. Most importantly, survivors’ mental health status needs to be considered as a priority concern to decrease the risk of further suicidality in the future.

## Figures and Tables

**Figure 1 ijerph-18-06326-f001:**
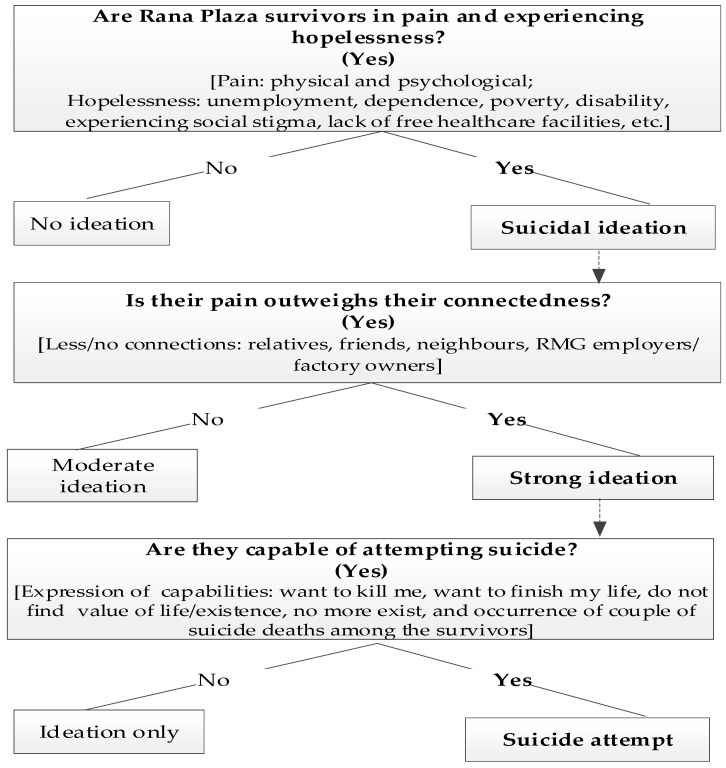
The three-step theory (3ST) of suicide (this modified version of the theory is quoted from Klonsky and May) [47].

**Table 1 ijerph-18-06326-t001:** Summary of themes.

Themes	Identified Issues
Gender, socio-economic, and cultural settings in shaping suicidal ideation	Gendered division of labour Socio-cultural construction of work Struggling with ongoing poverty Lack of access to RMG sector Unemployment Depression Losing hope to survive Anticipate finishing life
Ongoing physical injuries/disability, and mental trauma linked to suicidal ideation	Living with fear of amputating bodily parts, even after 6 years of the collapse Ongoing bodily pain, headaches, bone injuries/fractures, other physical illnesses sourced from the collapse Sleep disorders Lack of free treatment facilities Fear of being divorced due to the disability status Lack of connection with family and friends Life becomes meaningless/no value of existence Trauma
The feeling of ‘being a burden’ linked to suicidal ideation	Transformation of the status from breadwinner to bread eater Stress (reliance on family members for food and medicines) Intend to end life to be free from ‘being a burden’
Lack of social support and social stigma leading to suicidal ideation	Treated as being cursed Treated as *ALOKKHI* (a euphemism for loss/bad luck) Hiding the identity of being the Rana Plaza collapse survivors Misunderstanding and gap with the family members and relatives

## Data Availability

Data sharing is not applicable to this article.

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
