# Peer review of "Prevalence of Suicide Thoughts and Behaviours among Female Garment Workers Who Survived the Rana Plaza Collapse: An In-Depth Inquiry"

_ijerph, 2021, doi:10.3390/ijerph18126326_

Round 1
Reviewer 1 Report
Introduction: The Figure and discussion about 3ST needs to be described at the beginning of the paper and presented in more detail as part of the methods used.
References would be useful for the assertions about the patriarchal nature of the society and about the gendered nature of work.
A few edits:
Page 2: line 53 "unable to move thoroughly" Do you mean the woman was disabled?
Page 4: line 152: "were only one". I assume you mean was only one.
Author Response
Answer to reviewer
The authors are very grateful to the reviewer and to the editor for their suggestions to improve the quality of the article. Please see the point-by-point responses to the comments (attached). All authors contributed to the revision of the manuscript.
Humayun Kabir, PhD Student, BSS (Hons.), MSS, MA;
Professor Myfanwy Maple, PhD, GCTE, GradCertAdolHlthWelf, BSW (Hons1), GAICD;
Professor Kim Usher, RN, RPN, A/DipNEd, BA, BHSc, MNSt, PhD, FACN, FACM; &
Dr. Md Shahidul Islam, PhD, MSC, MSS, BSS (Hons.), DTMH, AFACHSM

Reviewer 2 Report
- Ln 53: delete "on her" at the end of the sentence.
- Expand the background section significantly, especially with regard to a) gender differences in suicidality and the role of gender norms in Bangladesh, b) the "lack of proper CSR initiatives" and c) literature regarding the experience of PTSD/psychological distress among disaster survivors.
- Ln 78: comma after "To do so"
- Explain how the subjects were recruited and the number of people in the "larger study" mentioned in Line 111. Were both males and females included in the "larger study?"
- The author needs to explain what is meant by "no sensitive information" being collected by the interviewer and why "no cultural issues arise." This is unclear.
- The authors need to expand the explanation of the data analysis. What was the start list of themes and how were these developed? Include a short review of the previous suicide-related qualitative studies that is mentioned.
- The first section of the results needs elaboration. What were the "physical, mental health conditions, and faced socio-cultural, and economic hardships" from which the participants were suffering? Were these pre-existing or were they a result of having survived the collapse?
- The second paragraph in the initial Results section should be rewritten to clarify that a set of themes were identified and will be individually analyzed. It currently reads like a conclusion rather than an introduction to the section.
- Ln 164: "sufferings" should be "suffering."
- Ln 234: comma after "sleeping"
- The last sentence of the first paragraph in the discussion mentions a distinction between "psychological pain" and "external factors." This needs elaboration. How are the different? Why is this important?
- Ln 324: clarify that the "previous studies" are about the collapse, specifically.
- The text in Figure 1 should be "Is their pain greater..." not "Are their pain greater..."
- It is not clear from the results how the researchers would answer the questions outlined in Figure 1. How do they know whether the individual is "capable" of suicide, for example? There is nothing in the results that speaks to that issue. And how does one quantify "pain" and "connectedness?" Are the participants asked specifically whether one outweighs the other? The model does not follow from the results provided.
- The conclusion is weak. There is no discussion of gender and no discussion of how these social supports could be provided. It is already well known that economic vulnerability, feelings of burdensomeness and stigma increase the risk of suicide. What does this study add? And what is specific to being women, other than the lack of other employment options? The authors acknowledge that the men who survived the collapse also suffer from lack of employment/economic concerns. Why did they choose to study women only? And why did they only interview women expressing thoughts of suicide instead of including women without ideation? They could have looked for differences among the suicidal group.
- This is an important topic but this paper needs much work.
Author Response

(The authors gave the same response as above.)

Reviewer 3 Report
The paper describes a study looking into the risk of suicidal ideation and behaviours in female survivors of the Rana Plaza collapse in 2013. Using a qualitative approach, interviews with female survivors indicated that these survivors were living with multiple risk factors that were directly linked to the Rana Plaza collapse, providing evidence for the proposition that lifelong care measures for these female survivors should be implemented to mitigate the risk of future suicide.
General judgment comments
The paper provides a clear introduction to the background of the Rana Plaza building collapse and suicide incidents involving survivors. Given the prevalence of traumatic experiences, investigating suicide risk factors in survivors of trauma is important for informing the implementation of measures in identifying and mitigating suicide risk in these survivors. In focusing on female survivors of the Rana Plaza building collapse, the paper discussed the unique sociocultural context of Bangladesh and explored its role in the development of suicide risk. Lastly, the paper also provides a perspective of suicide risk factors in the context of a low/middle income country (LMIC). Overall, some major revisions can be made to further strengthen the paper.
Major issues:
1) Thematic analysis
While the paper identified a number of core themes from the interviews, I think there are some areas that could be included:
(i) In the subsection 3.1 on socioeconomic settings, how does socioeconomic situation contribute to suicide risk? The subsection focuses more on the lack of employment opportunities (as opposed to socioeconomic situation per se) and it would be good to discuss how their socioeconomic situation contributes to suicide risk – eg. lack of financial ability to access both physical and mental health care, standard of living, quality of life
(ii) Considering that the discussion revolves around suicide ideation and behaviours, it seems that social support (or lack thereof, as suggested in the subsection on “being a burden”) should be discussed in the subsection 3.3. It is even cited in the discussion later that “losing friends/family and/or misunderstanding from them and social isolation are major causes of suicide ideation”, leading me to believe that a discussion on the social support systems of these female survivors is important as a suicide risk factor.
(iii) From the existing discussion, it seems like social stigma should be a subsection of its own. As suggested by the concept of “alokkhi” and hiding the fact that they are building collapse survivors, it seems that social stigma is contributing to their unemployment, strained relationships with family/neighbourhoods etc – all of which have been discussed and identified as suicide risk factors.
2) Implications
While the abstract raised the point that “In addition, a lifelong caring system (combining financial security and free healthcare) needs to be initiated to accommodate the female survivors with the mainstream society to avoid possible future suicide. They require long-term social and economic security and psychological support.”, this did not seem to be discussed at length (or at all) in the paper itself. It would be good to discuss the implications of the present findings in the discussion – for example, (i) in terms of healthcare and/or support measures that can be implemented to safeguard Rana Plaza survivors and (ii) what present findings suggest about suicide markers to look out for during follow up in survivors of similar or other types of complex trauma.
3) 3ST model
In raising the 3ST model, the paper provided Figure 1 in demonstrating the escalation towards a suicide attempt and “[incorporating] our findings”. However, it was not immediately clear how findings from the study was incorporated in the model.
A couple of suggestions to improve the existing figure for a better integration of findings:
(i) Label each step accordingly with a header as discussed in the text (eg. Step 1: Internalised hopelessness and pain). The question “Are Rana Plaza survivors in pain and [experiencing hopelessness]?” can then be provided.
(ii) Under each step, quotes and/or examples from existing findings can be included to demonstrate which of these findings fall under each category.
In addition, it was not immediately clear that “The step in moving from hopelessness into active intent to end one’s life has been the focus of much attention among suicidologists.” – It might be helpful for readers to clearly signpost this as “the third step”.
Minor issues:
1) It would be good to provide statistics of deaths by suicide amongst survivors (by gender, if possible) in the introduction – I believe this would help to better highlight the issue.
2) Visualisation of thematic analysis of interview
It might be helpful to have a table or diagram that illustrates the thematic analysis of the interview findings.
3) Limitations – On top of the small sample size, I believe that the study was limited to only looking at female survivors. Interviewing male survivors can also help to support findings in the female group by looking at similarities and differences across the theme identified in both groups.
Author Response

(The authors gave the same response as above.)

Round 2
Reviewer 2 Report
This is a much improved version. The addition of the table illustrating the themes for analysis is extremely helpful and the explanations of Bangladesh culture and gender norms will make the paper more relevant for international readers. I recommend one further minor revision: address the issue of external factors as a cause of suicide in the conclusion. The authors added a section earlier in the paper distinguishing between psychological factors (commonly assumed to be causal in Western societies) and external factors like life circumstances, which may be the central cause for suicide. This is an important distinction and critique of the way Western societies assume internal causation and ignore external factors. The conclusion to this paper would be stronger if it highlighted that point.
Author Response
Thank you very much for your valuable comments and suggestions. Please find the response attached.

Reviewer 3 Report
No suggestions
Author Response
Thank you!